# Villagers' Perceptions of Tourism Activities in Iona National Park: Locality as a Key Factor in Planning for Sustainability

**Jorge Morais [1], Rui Alexandre Castanho [2,3,4,5,6], Luis Loures [3,4,7,8] , Carlos Pinto-Gomes [1,9] and Pedro Santos [1,9,*]**

[1] Department of Landscape, Environment and Planning, School of Science and Technology, University of Évora, 7000-671 Évora, Portugal
[2] Faculty of Applied Sciences, WSB University, 41-300 Dąbrowa Górnicza, Poland
[3] Institute of Research on Territorial Governance and Inter-Organizational Cooperation, 41-300 Dabrowa Górnicza, Poland
[4] VALORIZA—Research Centre for Endogenous Resource Valorization, 7300 Portalegre, Portugal
[5] Environmental Resources Analysis Research Group (ARAM). University of Extremadura, 06071 Badajoz, Spain
[6] CITUR—Madeira-Centre for Tourism Research, Development and Innovation, University of Madeira, 9020-105 Funchal, Portugal
[7] Polytechnic Institute of Portalegre (IPP), 7300 Portalegre, Portugal
[8] Research Centre for Tourism, Sustainability and Well-being (CinTurs), University of Algarve, 8005-139 Faro, Portugal
[9] Institute of Mediterranean Agricultural and Environmental Sciences (ICAAM), University of Évora, 7000-671 Évora, Portugal
* Correspondence: aps@uevora.pt

**Abstract:** Local people's perception of nature-based tourism evolution and consequent impacts on their well-being are crucial in promoting ecotourism and achieving sustainable development. This study focused on indigenous populations' attitudes concerning tourism activities taking place in the Iona National Park, located in the Namibe Province of Angola, where ecotourism is considered an anchor product and is expected to become an economic driver of major importance. To obtain information that is useful for a changing tourism management in order to increase rural communities' well-being, we conducted a survey addressing local people's perceptions about the following main topics: perception of the presence of visitors and of their activities; present and prospective positive and negative effects of the park; and government policies that should be developed in the near future. The research shows that most respondents' perceptions strongly vary according to locality. The level of urbanization and the distance to the core areas of touristic activities appear to be the main factors driving the villagers' perception polarization. Thus, conflicting interests between nature conservation and local populations' well-being should be addressed and managed according to locality profiles, with some exceptions, such as ensuring the entire population has access to pastoral lands.

**Keywords:** ecotourism; destination management; sustainable development; Namibe province; Angola; Herero people

---

## 1. Introduction

Besides having well-known negative ecological impacts, the tourism industry may also negatively affect local people and their lifestyles, but such social and cultural effects are often overlooked by both visitors and hosts. Ecotourism has somehow emerged as a consequence of the growing dissatisfaction

with conventional forms of tourism that tend to disregard local social and cultural elements [1]. In fact, according to most definitions, ecotourism is the travel to areas that have high patrimonial values with the purpose of benefiting the local population, amongst other reasons. Securing the well-being of residents and of people who live nearby implies respect for their needs, their culture, and their relationship to the land. However, achieving this goal may prove difficult, especially in areas where residents are expected to forgo the use of renewable natural resources for other purposes. Frequently, different forms of tourism that support the aesthetic and recreational values of wildlife, even if they only involve non-consumptive uses, negatively affect the "residents' homes" by reducing food, water, and available space. Most wildlife management is habitat management, and increasing habitat quality for wildlife may imply diminishing pastoral areas or a decline in the carrying capacity for livestock grazing. Such human–wildlife conflict may not be expected to decrease unless adequate solutions to limit crop and livestock damages are developed.

Moreover, nature-based tourism also brings a certain degree of urbanization that may lead to further resource conflicts, which, if not promptly resolved, are likely to sow the seeds of socio-cultural problems [2]. In fact, collision between tradition and modernity, which is mainly a socio-cultural clash, has been pointed out as the most widespread and dangerous form of conflict in the newly built world [3,4]. Thus, decision makers engaged in promoting sustainable development have to consider simultaneously the implications of the economic, environmental, social, and cultural factors involved. The growing acceptance that decision-making practices must also consider socio-cultural issues, which ought to be monitored throughout the entire supply chain, has led to the emergence of methodological frameworks for integrated sustainability assessment (FISA), whose outputs should simultaneously capture the economic, environmental, and social impacts [5]. Planning for sustainable tourism, which also means planning for sustainable development, requires the use of these kinds of tools [6].

The purpose of this study is to provide the supportive information needed to change tourism management in Iona National Park (alternatively referred to as the Park) for the sake of conservation and sustainable development. A cardinal goal of conservation is "to ensure the greatest good for the most people over the long run" [7], and sustainable development is defined as "development that meets the needs of the present without compromising the ability of future generations to meet their own needs" [8]. Therefore, our intent is to contribute to mitigating current and latent conflicts of interest between tourism activities and the indigenous population in Iona National Park. For this purpose, we seek to disclose the villagers' perceptions of the effects of specific tourism activities and of the connected actions, practices, or policies on their resources needs, their values, and their relationships. Since long-term benefits for the local population is one of the main principles of ecotourism, future resource regulation in Iona National Park should favor indigenous interests. In order to contribute information that is useful to construct sound rationales in support of future tourism development decisions, we conducted a survey to address the local population's perceptions of four main topics: (1) perception of the presence of visitors and of their activities; (2) present and prospective positive effects of the park; (3) present and prospective negative effects of the park; and (4) government policies that should be developed in the near future. The respondents were categorized by gender, age-class, ethnic group, and locality. We hypothesized that locality would be the main factor that affected villagers' perceptions because human settlements within the park present significant differences regarding levels of urbanization and tourism development, and they are also very divergent with regard to elements of tourism attractiveness, both biotic and abiotic. We also analyzed whether age, gender, and ethnic group—demographic factors which are known to affect peoples' attitudes and opinions—influenced villagers' perceptions. In summary, the main goal of the present study is to provide Iona National Park authorities with information that may be useful to attain a sound compromise between two main principles of ecotourism: the well-being of the local population, and nature conservation. To make it easier for authorities to reach their recipients, we focused our analysis on finding differences between demographic profiles. We used descriptive statistics to summarize the answers, exploratory data analyses to better understand the structure of the data set, and inferential statistics to identify

statistically significant differences between the different categories of respondents. The identified patterns, relationships, and connections were then discussed in order to stress their underlying causes. In conclusion, suggestions for promoting sustainable tourism in Iona National Park and sustainable development in the Namibe Province are made.

*Literature Review*

Rapid population growth, pollution, excessive consumption of resources, and gradual deterioration of the land has led to an unprecedented degradation of the natural environment, and after two centuries of industrialization, human beings still do not know how to manage nature. Despite the great advances achieved in nature conservation in the last three decades, according to the Living Planet Report [9], Earth's resources are still being used faster than they can be renewed, and our impact on the planet has more than tripled since 1961. To help countries and communities conserve their genetic resources, species, ecosystems, and ecological processes, a network of protected areas has been put in place under the supervision of the International Union for the Conservation of Nature (IUCN), which is "dedicated and managed, through legal or other means, to achieving the long-term conservation of nature with associated ecosystem services and cultural values" [10].

Iona National Park, which is the oldest and largest Protected Area in Angola, covers around 15.150 km$^2$ and is located in the Namibe Province [11]. Established in 1937 as a game reserve, it was proclaimed a national park in 1964 under the Portuguese administration. The Angolan Civil War (1975–2002) led to a long period of abandonment, resulting in devastating consequences for wildlife. Several emblematic species became critically endangered or even extinct at the regional level, such as the black rhinoceros (*Diceros bicornis*), the African wild dog (*Lycaon pictus*), the black-faced impala (*Aepyceros melampus petersi*), the mountain zebra (*Equus zebra*), the African buffalo (*Syncerus caffer*), and the cheetah (*Acinonyx jubatus*). To reverse this situation, several organizations across Southern Africa have recently joined efforts, such as is the case of The Range-Wide Conservation Program for Cheetah and African Wild Dogs. Governmental authorities of Angola and Namibia work together to preserve a continuous block of the Namib Desert coastline and adjacent dunes, creating one of the largest transboundary conservation and tourism areas in Africa. Under the auspices of the United Nations, an Angolan National Project (United Nations Development Programme - Project ID 4082) aimed at conserving the country's biodiversity, beginning with the conservation of Iona National Park, was developed between February 2013 and April 2018. This project allowed the formation and specialized training of a "cross-functional team" and the rehabilitation of fundamental infrastructures. Moreover, it provided a zoning scheme to designate the sectors available for tourism purposes, set the basis for a cooperative experimental model of governance, and conducted an "Integrated Management Plan for the Iona National Park for the period 2015–2025" [12]. According to this Plan, the key to accomplishing the conservation goals of Iona National Park is to prevent resource use conflicts by putting in place solutions that safeguard the interests of local populations. Finding new solutions, or improving existing ones, is a continuous process to which we hope this work will contribute.

According to the Angolan Tourism Law, promulgated in 2015, tourism activities taking place in Protected Areas, such as Iona National Park, are termed "nature tourism", whereas "ecotourism", which is considered an anchor product in the Namibe's Tourism Master Plan [13] must comply with the following three principles: (1) is interested in natural and cultural areas; (2) contributes to nature conservation; and (3) benefits the local population. The term "ecotourism" emerged and began to appear in academic literature only in the late 1980s [14,15]. By the end of the last century, both the tourism industry and the academic community were expecting ecotourism to grow rapidly over the next two decades [16]. Official numbers provided by reference organizations over recent years confirmed that prediction [17]. In fact, according to reports regularly issued by the United Nations World Tourism Organization, nature tourism and ecotourism have been among the fastest growing sectors in the tourism industry. In Africa, nature-based tourism and ecotourism have already attained great relevance in countries such as South Africa and Tanzania, where these forms of alternative

tourism are associated with many prime sectors of those nations' economies, and they generate significant revenues that play an important role in funding conservation [18]. Although ecotourism is considered to be an effective development tool in many African nations [19–22], it is broadly acknowledged that its promotion in the vast areas of Southern Africa has been in the absence of guidelines specifically developed for the region [20]. Those guidelines should be developed, bearing in mind that indigenous populations have the most at stake. Ecotourism principles cannot be met, nor can sustainable development be achieved, unless local communities know in advance and formally accept the possible consequences of tourism development. Studies of rural communities in developing countries have found that access to conservation-related benefits and involvement of local people in decision-making for resource management can positively influence local attitudes towards wildlife, protected areas, and conservation [23–26]. Nevertheless, guidelines, as well as the research supporting them, have been mainly focused on the fair distribution of socioeconomic benefits rather than on the level of control that the local population should retain in areas they inhabit and are used to managing alone [27–29]. In protected areas where large herbivores and carnivores are abundant, the damages caused by wildlife on crops, pastures, and livestock are the main sources of dispute between local communities and conservation authorities for management control [30]. Moreover, this sort of human–wildlife conflict has been pointed out as a main impediment to ecotourism development in African countries [31]. To help overcome these differences, community-based natural resource management programs (CBNRMPs) have recently been created. In northern Namibia, a region near our study area that has been inhabited for centuries by the Herero people, a CBNRMP emerged in the late 1990s [32,33]. The proposal of a new management approach concerning this CBNRMP, which places more emphasis on collaboration between conservation agencies and indigenous populations, represents a further step in the reconciliation of biodiversity conservation goals and locals' land use rights and benefits [34].

The present literature review has shown that planning nature-based tourism in protected areas must be supported by the knowledge of local feelings toward the tourism industry at different levels. However, strategies and measures that work well in one place could be totally unsuitable somewhere else, and solutions people find appropriate in one place may not be applicable in another [35]. Thus, villagers' perceptions of tourism activities in Iona National Park are essential for planning ecotourism in this protected area.

## 2. Materials and Methods

### 2.1. Study Area

#### 2.1.1. The Villages

The research was conducted in seven villages located in the Namibe province, southwest Angola: Curoca, Espinheira, Garotas Novas, Iona, Monte Negro, Ngulova, and Pediva. All of these villages are located between the Curoca and Cunene Rivers and are spread over a vast area, as shown in Figure 1. The villages differ widely from one another regarding geographic location, infrastructure, social facilities, administrative relevance, number of villagers, and ethnic composition of the population.

This plurality of communities is representative of the diversity of human settlements that exist in the study area. Iona is the largest village; it is a commune headquarters, has a primary school, a health post, a police station, and presents a fast-growing population. Monte Negro and Garotas Novas are two smaller villages that also belong to Iona's commune. In both villages, many basic services are lacking, and their population is mainly composed of nomadic pastoralists. Ngulova is a relatively large community, consisting of numerous family units, but with regard to its infrastructures and social facilities, it is on the opposite side of Iona and has no administrative relevance. Most of its population is composed of nomadic pastoralists who own livestock and buy goods in the streets from itinerant traders. Pediva is mostly known for its hot sulphuric water springs, which have formed a small lake in the desert. Besides basic tourism facilities, such as a camping park, the village has a primary school

and was recently armed with a police station. Espinheira is located in the heart of Iona National Park, and it acts as a central administrative base that provides support for tourism activities and logistics management. For Curoca, which is near Tômboa city, there are plans to build infrastructure and facilities in order to increase the region's tourism carrying capacity.

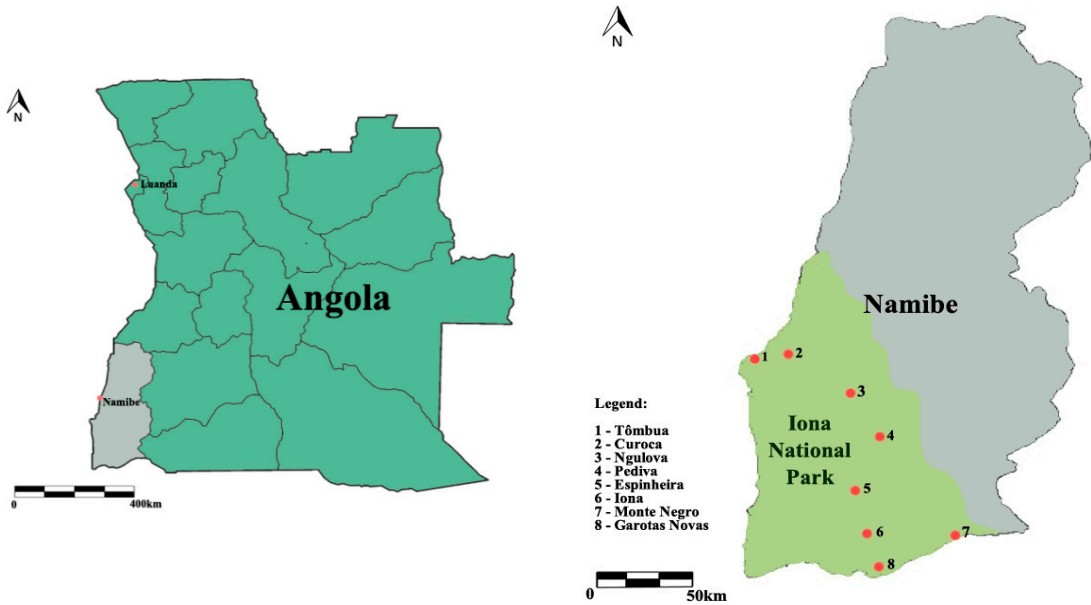

**Figure 1.** The study area includes seven villages—Curoca, Ngulova, Pediva, Espinheira, Iona, Monte Negro, and Garotas Novas—that represent the diversity of human settlements existing in Iona National Park, a protected area located in Namibe Province (southwest Angola).

### 2.1.2. The People

The study area is mainly inhabited by pastoral or agro-pastoral populations living in rural communities in which animal husbandry has central social and cultural roles [36,37]. Dairy products are of major importance in the populations' diet, and the need for a cattle enclosure near dwellings leads to a relatively scattered settlement pattern. Low rainfall and drought are perhaps the greatest environmental problems, and nomadism or semi-nomadism are ecological adaptations to such harsh conditions [38]. Whenever the amount and regularity of rainfall assures viable farming, women cultivate small plots, mainly for self-consumption.

With the exception of a few "pre-Bantu" groups, these populations are Bantu-speaking, and according to the "ethnic map of Angola", belong predominantly to the Herero ethno-linguistic group [39]. A complex and dynamic prehistory led to a subethnic segregation that was accentuated by successive migrations that resulted from the colonization process [34]. Among the Herero group, we have found people that can be identified with one of the following groups: Cuanhoca, Cuepe, Cuísse, Cuvale, Himba, Mucubal, Mundimba, and Mwakahona. These group designations were established mainly in the twentieth century during the Portuguese administration. This ethnic classification may be questionable [39], but today those groups are widely recognized and their names well-accepted. Quimbar is another group present in the study area. With roots in the Herero people, this group was formed by slave descendants that mixed with foreign people and assimilated into the Portuguese culture. The region is also inhabited by people belonging to the largest ethno-linguistic group in Angola, the Ovimbundu, whose origins lie outside the Namibe Province.

### 2.2. Data Collection and Analysis

Villagers inhabiting rural communities located both inside Iona National Park and in surrounding areas make up the target population (approximately 2300 people) [12]. In order to get a representative

sample of the above-mentioned diversity of communities, the number of respondents selected in each village was approximately proportional to the community population size, including resident and itinerant people. Within each village, the respondents were selected by haphazard (convenience) sampling. Haphazard sampling was the only possible option since there were no sampling frames (i.e., lists of all of the groups inhabiting each village), communities extended over loosely defined areas, and the populations' demographic structure and ethnical distribution were unknown. The questionnaires were administered to people who were easy to contact or to reach, and the only sampling criterion was willingness to answer the questions. Thus, our sample may not be fully representative of the total population of the Park, but it allows us to make scientific generalizations about the villagers who were easy to locate and willing to respond, i.e., the members of the population that traditionally are the leaders and the opinion makers. Considering the respondents from all of the studied villages, the sample size—202 villagers who answered all questions—was large enough to allow statistic inferences. IBM Statistical Package for Social Sciences (SPSS) was used to conduct descriptive statistics, including multiple correspondence analyses, and inferential statistics.

*2.3. Survey Design and Administration*

The survey included two different sets of variables: variables concerning respondents' demographic profile and variables concerning respondents' perceptions about nature-based tourism in Iona National Park. The following demographic profile variables, also known as biodata, were analyzed: locality, ethnic group, gender, and age. For these four variables, the response options were exhaustive and mutually exclusive. As for the villagers' perceptions about nature-based tourism in Iona National Park, four topics were considered: (1) perception of the presence of visitors and of their activities; (2) present and prospective positive effects of the Park; (3) present and prospective negative effects of the Park; and (4) government policies that should be developed in the near future. To avoid double-barreled questions, these three matters were deployed in different items.

To address the first topic, the following nine questions were asked: (a) Has your locality been visited by tourists recently? (b) Are most tourists domestic or foreign? (c) Do tourists fish? (d) Do tourists hunt? (e) Do tourists watch wildlife? (f) Do tourists watch plants? (g) Do tourists enjoy the landscape? (h) Do tourists take pictures? (i) Do tourists speak with the local population?

The second topic covered different items, which were separately addressed by asking the following six questions: (a) Are locals in favor of the Park's development? (b) Does the Park create direct jobs? (c) Does the Park create indirect jobs? (d) Does the Park help to value the customs and traditions of local populations? (e) Does the Park help to recover the fauna and flora? (f) Does the Park contribute to improving access to health care and education?

The third topic also covered several different items, separately addressed by asking the following seven questions: (a) Does the Park contribute to the breakdown of family relationships within the community? (b) Does the Park contribute to the disturbance of the tranquility of the population or of their cultural manifestations? (c) Does the Park contribute to the building of infrastructure not wanted by the population? (d) Does the Park contribute to increased garbage produced by tourists? (e) Does the Park limit hunting and fishing? (f) Does the Park limit pastoral activities? (g) Does the Park limit the collection of firewood?

The fourth topic was addressed by asking the following five questions: (a) Should the government improve motor vehicle traffic in the Park? (b) Should the government create new structures to accommodate tourists? (c) Should the government build a national road connecting the Park to the neighboring Namibian Republic? (d) Should the government repopulate the Park with its emblematic species? (e) Should the government promote and implement ecotourism?

All were closed-ended questions to be answered with "yes", "no", or "do not know/no answer". We deliberately chose closed-ended questions because of the expected communication problems, mainly because many respondents had difficulty with speaking Portuguese. Some respondents were

only proficient in their own geographic dialect. Additionally, closed-ended questions facilitated the categorization of respondents and allowed us to conduct statistically significant tests.

The questionnaires were administered from April to June 2017 in person. We used this this type of interview because it allowed the respondents the opportunity to ask for further clarification during the conversation [40–43]. The guards of the park acted as interviewers, using interpreters whenever necessary. Responses were manually transferred from paper questionnaires to a spreadsheet. Numbers or "codes" were then assigned to each possible answer and manually entered into the spreadsheet while going through each respondent's questionnaire.

## 3. Results

### 3.1. Respondents' Demographic Profile

The population was stratified by locality to proportionally represent the different villages. Thus, as can be seen in Table 1, almost 70% of respondents belonged to the two most populated areas, i.e., Iona and Ngulova, whereas only 3% were interviewed in Espinheira, a small locality that corresponds to an administrative base. Admitting that each individual, regardless of ethnicity, had the same probability of been sampled, a proportional representation of the different ethnic groups inhabiting the study area was obtained (see Table 1). The results showed that the variable "ethnic group" was not independent of "locality" (Fisher's exact test = 155.227, $p < 0.001$), and consequently, the predominant ethnic groups in Iona (Himba) and in Ngulova (Cuísse and Cuvale) constituted almost half of the respondents. If we assemble the ethnic groups Cuanhoca, Cuepe, Cuísse, and Cuvale together, the Himba and Mundimba ethnicities together, and assemble the remaining ethnic groups in a third cluster, we may see that localities' ethnical distributions are framed by a geographic pattern, as shown in Figure 2.

**Table 1.** Distribution of respondents regarding demographic characteristics. The percentage of respondents belonging to each variable category is shown in parentheses.

| Locality | | Ethnic Group | |
|---|---|---|---|
| Curoca | 7 (3.5%) | Cuanhoca | 18 (8.9%) |
| Espinheira | 6 (3.0%) | Cuepe | 22 (10.9%) |
| Garotas Novas | 14 (6.9%) | Cuísse | 29 (14.4%) |
| Iona | 90 (44.6%) | Cuvale | 20 (9.9%) |
| Monte Negro | 23 (11.4%) | Himba | 46 (22.8%) |
| Ngulova | 43 (21.3%) | Quimbar | 26 (12.9%) |
| Pediva | 19 (9.4%) | Mucubal | 16 (7.9%) |
| | | Mundimba | 9 (4.5%) |
| | | Mwakahona | 4 (2.0%) |
| | | Ovimbundu | 12 (5.9%) |
| **Gender** | | **Age** | |
| Men | 131 (64.9%) | < 3 0 years | 14 (6.9%) |
| Women | 71 (35.1%) | 30–40 years | 66 (32.7%) |
| | | 41–50 years | 102 (50.5%) |
| | | > 50 years | 20 (9.9%) |

In northern localities of the study area, such as Curoca and Ngluva, there was a predominance of the first cluster's ethnicities, while in southern localities of the study area, such as Garotas Novas and Monte Negro, predominant ethnic groups were from the second cluster. In Iona, being the commune headquarters, all ethnicities seemed to be well represented, and in localities more associated with tourism activities, such as Espinheira and Pediva, there was a predominance of the remaining ethnic groups, in particular Quimbar and Ovimbundu. However, when controlling for locality, for example by separately analyzing the results referring only to Iona, no significant differences among ethnic groups were found regarding any of the studied topics.

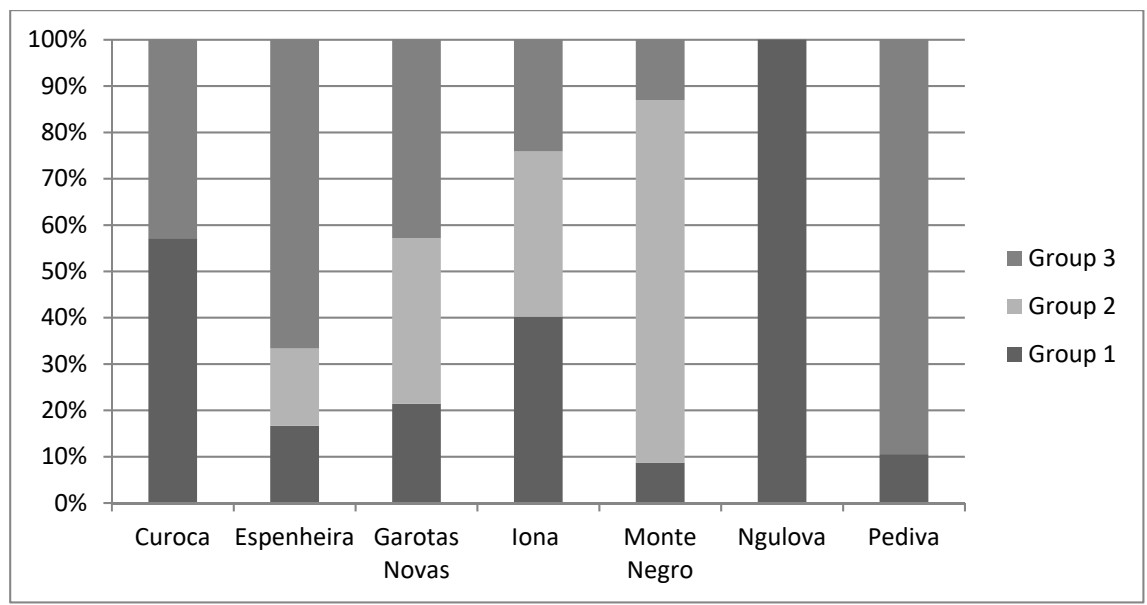

**Figure 2.** Ethnicity distribution by locality considering the following groups: Group 1—Cuanhoca, Cuepe, Cuísse, and Cuvale; Group 2—Himba and Mundimba; and Group 3—Mucubal, Mwakahona, Quimbar, and Ovimbundu.

### 3.2. Perception of the Presence of Visitors and of Their Activities

Table 2 shows that all respondents were aware of the presence of visitors in the study area, since they unanimously answered "yes" to the question "Has your locality been visited by tourists recently?" However, villagers' perceptions varied according to locality in what concerns tourist origins (Fisher's exact test = 17.905, *p* = 0.003). According to villagers' perceptions, the vast majority of visitors to Iona National Pak appreciated activities that were not specific to nature-based tourism destinations, such as taking pictures, speaking with the local population, and enjoying the landscape (see Table 2). Regarding activities specific to a nature-based tourism context, respondents' perceptions completely diverged depending on whether or not tourism activities involved consumptive recreation. Non-consumptive activities, such as watching plants and wildlife, were perceived as tourism activities by many villagers, but just a few respondents had seen visitors fishing, and none of them had seen visitors hunting (see Table 2). Villagers' responses were repeatedly consistent, suggesting that respondents tended to give reliable answers.

**Table 2.** Distribution of respondents regarding perception of tourism activities in the Iona National Park. The percentage of respondents belonging to each variable category is shown in parentheses.

| Asked Questions | Yes | No | Do not Know/No Answer |
|---|---|---|---|
| Has your locality been visited by tourists recently? | 202 (100%) | 0 (0%) | 0 (0%) |
| Are most tourists domestic or foreign? | 46 (22.8%) | 156 (77.2%) | 0 (0%) |
| Do tourists fish? | 18 (8.9%) | 133 (65.8%) | 51 (25.2%) |
| Do tourists hunt? | 0 (0%) | 1 (0.5%) | 201 (99.5%) |
| Do tourists watch wildlife? | 148 (73.3%) | 0 (0%) | 54 (26.7%) |
| Do tourists watch plants? | 83 (41.1%) | 0 (0%) | 119 (58.9%) |
| Do tourists enjoy the landscape? | 178 (88.1%) | 0 (0%) | 24 (11.9%) |
| Do tourists take pictures? | 201 (99.5%) | 0 (0%) | 1 (0.5%) |
| Do tourists speak with the local population? | 199 (98.5%) | 0 (0%) | 3 (1.5%) |

### 3.3. Present and Prospective Positive Effects of the Park

A large majority of the respondents considered that the local population was in favor of the Park's development and also agreed that it may well bring considerable benefits to their communities, as shown in Table 3. Almost no unfavorable opinions were expressed regarding this topic; the exceptions

were two "No" answers to the question "Does the Park create direct jobs?", but with respect to many questions, a significant part of the respondents showed some scepticism by choosing the option "Do not know/No answer" (see Table 3).

**Table 3.** Distribution of respondents regarding perception of present and prospective positive effects of Iona National Park. The percentage of respondents belonging to each variable category is shown in parentheses.

| Asked Questions | Yes | No | Do not know/No Answer |
|---|---|---|---|
| Are locals in favor of the Park development? | 158 (78.2%) | 0 (0%) | 44 (21.8%) |
| Does the Park create direct jobs? | 144 (71.3%) | 2 (1.0%) | 56 (27.7%) |
| Does the Park create indirect jobs? | 135 (66.8%) | 0 (0%) | 67 (33.2%) |
| Does the Park help to value the customs and traditions of local populations? | 158 (78.2%) | 0 (0%) | 44 (21.8%) |
| Does the Park help to recover the fauna and flora? | 190 (94.1%) | 0 (0%) | 12 (5.9%) |
| Does the Park contribute to improving the access to health care and education? | 180 (89.1%) | 0 (0%) | 22 (10.9%) |

## 3.4. Present and Prospective Negative Effects of the Park

Few respondents considered that the Park negatively affected family relationships, population tranquility, and their cultural manifestations, or that it contributed to the building of infrastructures not wanted by local people; the data are shown in Table 4. In fact, the answers to the first three questions of this topic reflected either a positive perspective, given by those who chose the option "No", or a prudent skepticism, revealed by those who chose the option "Do not know/No answer". According to the large majority of respondents, the Park contributed to increased garbage production, and it limited hunting and fishing, as well as pastoralism and firewood collection (see Table 4).

**Table 4.** Distribution of respondents regarding perception of present and prospective negative effects of Iona National Park. The percentage of respondents belonging to each variable category is shown in parentheses.

| Asked Questions | Yes | No | Do not Know/No Answer |
|---|---|---|---|
| Does the Park contribute to the breakdown of family relationships or within the community? | 12 (5.9%) | 87 (43.1%) | 103 (51.0%) |
| Does the Park contribute to the disturbance of the tranquility of the populations or their cultural manifestations? | 26 (12.9%) | 84 (41.6%) | 92 (45.5%) |
| Does the Park contribute to the construction of infrastructures not wanted by the population? | 31 (15.3%) | 93 (46.0%) | 78 (38.6%) |
| Does the Park contribute to increased garbage produced by tourists? | 152 (75.2%) | 31 (15.3%) | 19 (9.4%) |
| Does the Park limit hunting and fishing? | 187 (92.6%) | 0 (0%) | 15 (7.4%) |
| Does the Park limit livestock activities? | 202 (100%) | 0 (0%) | 0 (0%) |
| Does the Park limit the collection of firewood? | 198 (98.0%) | 0 (0%) | 4 (2.0%) |

## 3.5. Government Policies That Should be Developed in the Near Future

Table 5 shows that most respondents seemed to support possible government policies to be implemented in the near future that would favor motor vehicle traffic within the Park, the building of touristic infrastructures, and the development of ecotourism. In Pediva and Espinheira, two villages where tourism already plays a fundamental socio-economic role, the respondents were unanimously in favor of those policies. Respondents were divided about whether the government should build a national road connecting the Park to the neighboring Namibian Republic, but there was a broad consensus on the idea that the government should put in place policies intended to promote the recovery of the Park's emblematic species (see Table 5).

**Table 5.** Distribution of respondents regarding government policies that should be implemented in the near future. The percentage of respondents belonging to each variable category is shown in parentheses.

| Asked Questions | Yes | No | Do not Know/No Answer |
|---|---|---|---|
| Should the Government improve motor vehicle traffic within the Park? | 195 (96.5%) | 1 (0.5%) | 6 (3.0%) |
| Should the Government create new structures to accommodate tourists? | 197 (97.5%) | 0 (0%) | 5 (2.5%) |
| Should the Government construct a national road connecting the Park to neighboring Namibian Republic? | 67 (33.2%) | 40 (19.8%) | 95 (47.0%) |
| Should the Government repopulate the Park with its emblematic species? | 201 (99.5%) | 0 (0%) | 1 (0.5%) |
| Should the Government promote and implement ecotourism? | 184 (91.1%) | 0 (0%) | 18 (8.9%) |

The graphic results of the multiple correspondence analysis (MCA) used to uncover the relationships among the studied categorical variables are presented in Figures 3 and 4. The projection of respondents on the first plane, presented in Figure 3, showed that the respondents within the same locality were close together. The discrimination measures of the variables are exhibited in Figure 4, which revealed that locality presented the highest values in both dimensions. For Dimension 1, which is represented by the horizontal axis and accounts for 24.74% of the variance, we see in Figure 4 that respondents from Iona (e.g., object points 116, 133, 134, and 161) and respondents from Ngulova (e.g., object points 72, 75, 81, 86, and 93) were furthest away from the origin, and therefore had the most importance. Besides locality, the variables referring to the following questions also had high discrimination values: Does the Park contribute to the breakdown of family relationships or within the community? Does the Park contribute to the disturbance of the tranquility of the population or their cultural manifestations? Does the Park contribute to the construction of infrastructures not wanted by the population? Does the Park contribute to increased garbage produced by tourists? Thus, the divergence for Iona and Ngulova was mainly due to differences between these localities regarding villagers' perceptions concerning the negative aspects of the Park. For Dimension 2, which is represented by the vertical axis and accounts for 12.84% of the variance, we see that respondents from Garotas Novas (e.g., object points 6, 7, 12, 14, 15, and 16) and respondents from Espinheira (e.g., object 24, 25, 26, 27, 28, and 29) had the most importance. Besides locality, only the variable referring to the question "Do tourists fish?" presented a high value. These results indicate that the largest deviation from independence in the sample was between Iona and Ngulova, and seemed to result mainly from differences regarding levels of urbanization and tourism development. The second most important difference was between Garotas Novas and Espinheira, and seemed to result mainly from differences regarding tourism facilities and elements of attractiveness. All respondents from Curoca are close to the origin, which means that they tend to be neutral regarding both dimensions.

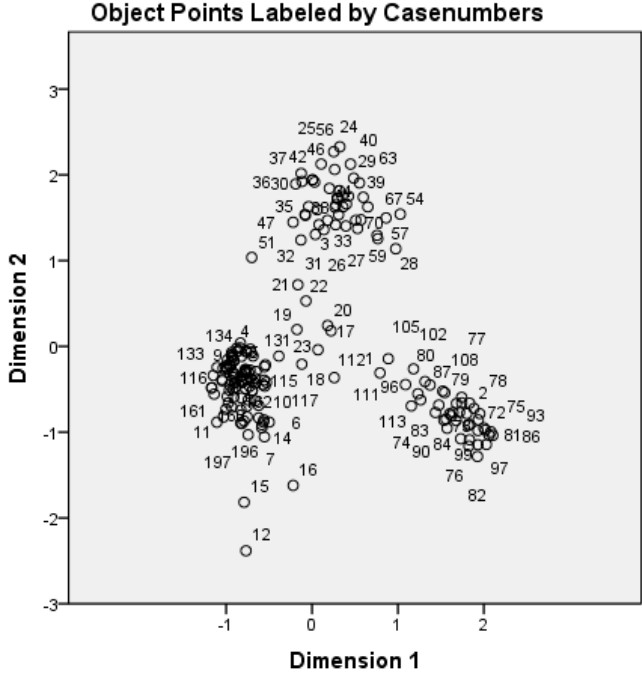

**Figure 3.** Projection of respondents (object points) on the first plane given by MCA. The distance between respondents, both along the horizontal and the vertical axes, gives a measure of their similarity (or dissimilarity). Respondents with similar profiles are close together on the map. The respondents are distributed by localities as follows: Curoca—1, 18–23; Espinheira—24–29; Garotas Novas—4–17; Iona—55, 114–202; Monte Negro—3, 30–51; Ngulova—2, 72–113; Pediva—52–54, 56–71.

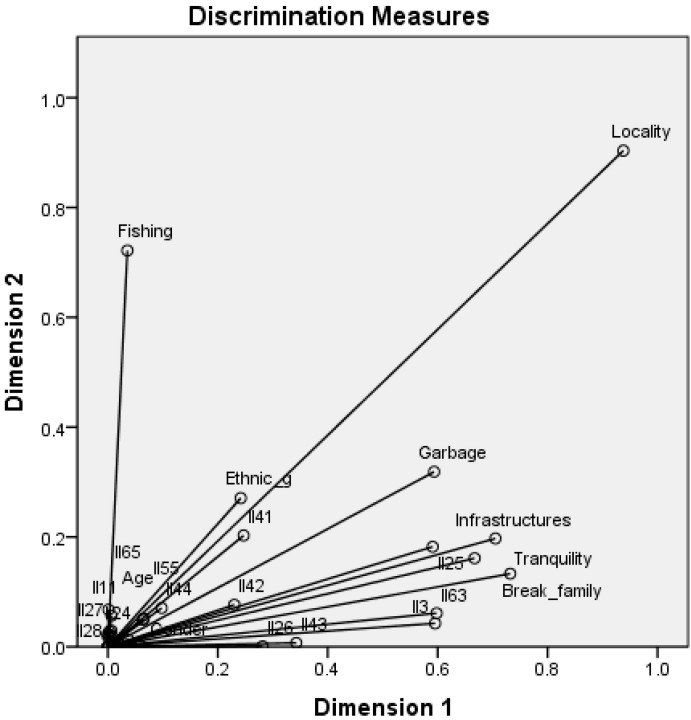

**Figure 4.** Discrimination measures of the MCA dimensions. Only the variables with the highest values are presented using words, the others are presented using their labels.

## 4. Discussion

The two first components or dimensions account for about 40% of the variance. These results suggest the use of an analytical approach to break down the complex system of relationships into smaller pieces. Herein, we analyze in detail some of those small pieces by comparing villagers' perceptions between the different classes of age, gender, ethnic group, and locality. We also analyze whether these demographic variables are independent of each other. Finding statistically significant differences between different demographic profiles would make it easier for the Park authorities to reach their intended recipients, and thus, to implement the desired measures and actions.

### 4.1. Respondents' Demographic Profile

Haphazard sampling conducted in each locality led to a selection bias regarding gender and age variables. Assuming that the demographic structure of the studied population is similar to that presented by the Angolan total population, our sample included a disproportionate number of males and a disproportionate number of villagers belonging to the age-class 41–50 years (see Table 1). Gender and locality were not independent variables (Fisher's exact test = 15.484, $p = 0.013$), and sampling bias regarding gender did not occur in all localities. The largest disparities were registered in Espinhera (0 females versus 6 males) and Ngulova (8 females versus 35 males), while in Iona the number of interviewed females was not significantly different from that expected if the sex ratio was 1:1 (Binomial test, $p < 0.001$). The strongly unbalanced sex-ratio verified in Espinheira resulted from the fact that this was not a residential village, but it was a working community where the main jobs are typically held by men. In Ngulova, the disproportionate number of males probably reflects the traditional society organization, since women work harder and longer hours than men [38,44]; men's traditional roles as pastoralists and village defenders make them less busy, and therefore, more available and willing to take part in the survey. Age and locality were independent variables (Fisher's exact test = 22.342, $p = 0.122$), and oversampling of elder villagers, mostly belonging to the age-class 41–50 years, was verified in all localities except Espinheira (due to its already mentioned specificity). This general pattern reflects these societies' social and political structures, in which elders, both men and women, are the traditional power holders to whom younger people refer their problems. Therefore, it is no surprise that elders took priority in the survey. As they are decision makers and opinion leaders, their perceptions are expected to sway younger generations, and thus, to a certain extent, they represent the entire population.

### 4.2. Perception of the Presence of Visitors and Their Activities

The strong presence of domestic tourists perceived by Pediva respondents is easily understandable and this was expected, as long ago the village was a popular thermal tourism destination, and today it is considered a classic stop among the domestic tourism routes. The perception in most villages that the majority of visitors are foreign tourists is in agreement with the official record of visitors to Iona National Park in 2015 from data collected by Angolan Foreign and Migration Service. These results revealed that, more than noticing the presence of visitors, the villagers were able to identify whether tourists were domestic or foreign.

The respondents' perceptions regarding non-consumptive nature-based tourism activities were fully in accordance with the characteristics of visitors to Iona National Park, who predominantly fit a "wildlife tourist" profile [45]. Furthermore, the interest of visitors in watching wildlife has been consistently identified in other similar African destinations as the main tourism attraction [18,46–48]. It is noteworthy that people from the Iona commune, which is located at the heart of the Park, were more aware of visitors interested in watching wildlife than, for instance, people of Ngulova (Fisher's exact test = 42.468, $p < 0.001$). On the other hand, it was in buffer areas around the heart of the Park, such as Curoca and Ngulova, that more respondents than was expected pointed out the visitors' interest in watching plants.

Even though hunting is a very common and profitable form of consumptive recreation in sub-Saharan Africa [49], no hunting tourism was detected in the study area by any of the respondents; cautiously, almost all selected the option "Dot not know/No answer" to answer the question "Do tourists hunt?", but no one chose the option "Yes". Such unanimous perception seems to be entirely accurate, since hunting is not allowed in the Park. As for tourism fishing, the vast majority of respondents appeared to have a well-defined opinion; most of them answered "No" to the question "Do tourists fish?", while only a few chose the option "Yes". However, differences were found regarding ethnic group and locality variables. In fact, sample data enable the null hypotheses, that the frequencies of these variables are equally distributed across their categories, to be rejected (ethnic group—Fisher's exact test = 42.468, $p < 0.001$; locality—Fisher's exact test = 266.045, $p < 0.001$). Garotas Novas stands out from all other villages, as clearly shown by the MCA, as the place of choice to practice tourism fishing—13 out of 14 respondents declared to be aware of this activity. On the other hand, in Garotas Novas the most numerous ethnic group was Quimbar; thus, the popularity of tourism fishing in the village may be explained by a favorable combination of good natural conditions, with the strong presence of a people accustomed to dealing with foreign tourists. Quimbar people, contrary to nomadic pastoralists, have gradually assimilated into Western culture.

*4.3. Present and Prospective Positive Effects of the Park*

Women appear to be less skeptical than men regarding the questions "Are locals in favor of the Park development?" ($\chi^2 = 7.104$, degrees of freedom = 1, $p = 0.008$), "Does the Park create direct jobs?" (Fisher's exact test = 5.161, $p = 0.049$), and "Does the Park contribute to improving the access to health care and education?" ($\chi^2 = 5.012$, d.f. = 1, $p = 0.025$). However, as gender and locality were not independent variables, these results must mainly reflect the perspective of women from Iona, as they represent about 60% of the total number of female respondents. As for ethnic group, northern ethnicities, such as Cuanhoca, Cuepe, Cuvale, and Cuísse, seemed to be more inclined to skepticism than southern ethnicities, such as Himba, with respect to the questions "Are locals in favor of the Park's development?" (Fisher's exact test = 35.247, $p < 0.001$), "Does the Park create direct jobs?" (Fisher's exact test = 34.707, $p = 0.002$), and "Does the Park help to value the customs and traditions of local populations?" (Fisher's exact test = 18.542, $p = 0.019$). Such geographical divergence of ethnicities is not surprising, since ethnic group and locality are not independent variables.

Locality emerged as a key factor behind the polarization of the respondents' skepticism. In fact, regarding all of the questions in this topic, the sample data enabled us to reject ($p < 0.001$) the null hypothesis that skepticism is equality distributed across the categories of the variable locality. People from Ngulova tended to be more skeptical than people from other communities. For instance, only a small minority of respondents from Ngulova (9 out of 43) chose the option "Yes" to answer the question "Are locals in favor of the Park's development?", while in Iona all 90 respondents answered "Yes" to the same question. As for the question "Does the Park create direct jobs?", far less than expected respondents answered "Yes" in Ngulova (expected count = 30.7; count = 12), while in Iona, opposing results were obtained, as more respondents than expected answered "Yes" to the same question (expected count = 64.2; count = 77). Overall, a very significant statistical difference was found amongst the studied localities regarding this question (Fisher's exact test = 71. 23, $p < 0.001$). Again, regarding indirect job opportunities created by the Park, answers varied widely across localities (Fisher's exact test = 55.334, $p < 0.001$), but the same opposition was verified between Ngluva and Iona, i.e., most respondents in Iona expressed a favorable opinion, whereas the majority of respondents in Ngulova revealed skepticism. A similar distribution pattern of answers was observed regarding the other questions included in this topic, except for the last one, "Does the Park contribute to improve the access to health and education?" For this particular question, Ngulova and Iona did not show opposite trends. Respondents from both of these localities agreed that the Park contributes to improving access to health and education. On this subject people from Garotas Novas seem to diverge from everybody

else, since only in this village the majority of the respondents (8 out of 14) answered that question with the option "Do not know/No answer".

*4.4. Present and Prospective Negative Effects of the Park*

The results varied according to the respondents' demographic profile; however, since gender and ethnic group were not independent from locality, we focused the analysis on locality and age, as the latter was the only demographic variable that was independent from locality. Once again, as clearly shown by the MCA, Ngulova and Iona stood on opposite sides. Regarding the question "Does the Park contribute to the breakdown of family relationships or within the community?", all respondents from Ngulova chose the option "No", showing a unanimous, positive impression, whereas Iona was the only village where none of the respondents chose the option "No", and the number of respondents who chose the option "Yes" was much higher than expected. Conversely to what happened with people from Ngulova, it seems that people from Iona either doubted the effect that the Park has on family relationships or considered it as pernicious. The same type of antagonism between Ngulova and Iona was verified in relation to the questions "Does the Park contribute to the disturbance of the tranquility of the population or their cultural manifestations?" and "Does the Park contribute to the building of infrastructures not wanted by the population?" Once more, respondents from Ngulova did not notice or anticipate negative effects on the well-being of local people regarding these two matters—all respondents chose the option "No" to answer the first question, and 39 out of 41 also answered "No" to the second question. Otherwise, most respondents from Iona did not take sides in these issues—79 out of 90 and 61 out of 90 chose the option "Do not know/No answer" to respond to the first and the second questions, respectively; the remaining respondents answered "Yes" to both questions. Elders, primarily belonging to the age-class > 50 years, were consistently the most unsure respondents, choosing the option "Do not know/No answer" more often than expected to respond to both questions. People from Ngulova did not perceive many of the positive effects mentioned in the previous topic nor the negative impacts referred to in the last three questions. Such a general lack of awareness regarding tourism effects, irrespective of their nature, suggests that tourism is seen by people from Ngulova as a distant reality that can hardly affect their village. Conversely, people from Iona generally acknowledge the positive effects of tourism in the Park, but on the other hand, also recognize that tourism may negatively affect their well-being. This divergence of perceptions appears to reflect the far-reaching differences between the two villages; Iona, located inside the Park and being a commune headquarters, offers more tourism-related job opportunities and socio-economic benefits than Ngulova, whose population is mostly composed of nomadic pastoralists.

Regarding the last four questions, the usual opposition between Ngulova and Iona was observed only for the question "Does the Park contribute to increased garbage produced by tourists?" Respondents from Ngulova chose the option "No" more often than expected (expected count = 6.6; count = 30) to answer this question, whereas all respondents from Iona answered "Yes" to the same question. It seems that tourism in Ngulova either takes place far from the village or the visitors avoid producing garbage, showing high environmental awareness. Conversely, most visitors to Iona appear to be high-impact tourists, and obviously, their profile does not fit any ecotourism definition [1,50]. Unanimously, the respondents believed that the Park limited the local population's access to pastoral land use, and with the exception of a few doubters mostly belonging to Garotas Novas and Iona, they tended to agree that the Park also limited the use of other natural resources, such as wildlife and firewood. This negative impression of the Park is common to many other indigenous communities, which as a consequence of state-imposed rules, were displaced from their traditional areas or had to cope with limited access to resources [34,51,52].

*4.5. Government Policies That Should be Developed in the Near Future*

Bearing in mind the benefits that cross-border cooperation (CBC) may offer [41,53], the answers to the question "Should the government build a national road connecting the Park to the neighboring

Namibian Republic?" were rather surprising. Opinions varied widely regarding age (Fisher's exact test = 22.492, $p = 0.001$) and locality (Fisher's exact test = 156.312, $p < 0.001$). The percentage of respondents that chose the option "Do not know/No answer" to respond to the question progressed steadily with age, increasing from less than 40% in the first two age-classes to 80% in the age-class > 50 years. Elders, who tend to be leaders of opinion and decision makers, were the most uncertain respondents, showing how many communities were unsure about this issue. In Pediva the option "Do not know/No answer" was not chosen by any respondent, but this community was also divided because the number of respondents that answered "Yes" and the number of those that answered "No" were very similar, with 10 and 9 respondents, respectively. Such strong polarization of opinions may reflect an underlying conflict of interest—the interests of those who do not want to lose domestic tourists and the interests of those who want to attract more foreign visitors. Ngulova, yet again, was the only exception, since clearly most villagers appeared to be against building roads; respondents chose the option "No" more than expected (expected count = 8.5; count = 29).

## 5. Conclusions

For the purpose of integrating tourism policies and native population's interests in the Namibe Province, Angola, a survey of villagers' perceptions of nature-based tourism activities was conducted in Iona National Park and surrounding areas. The results allow us to conclude that as hypothesized, many respondents' perceptions were strongly polarized based on locality, a variable that was not independent from ethnicity. The polarization pattern was common to most of the different subjects addressed, namely regarding the perceived positive and negative impacts of the Park, and concerning the government policies that should be developed in the near future. Two main factors appeared to promote polarization of the villagers' perceptions: the level of urbanization and the distance to core areas of touristic activities. According to these factors, Iona and Ngulova are villages situated at opposite poles, as the divergences between their dwellers' perceptions were extreme. The respondents from Iona, which is a commune headquarters located in the heart of the Park, were much more aware of both the positive and negative effects of the Park than the respondents from Ngulova, a rural village mainly inhabited by seminomadic pastoralists. As for government policies that should be developed in the near future, only in Ngulova area the majority of respondents declared to be against building a national road connecting the Park to the neighboring Namibian Republic. However, respondents from all localities tended to agree that the Park significantly limited the use of natural renewable resources by the local population, such as firewood, animals they hunt, and fish for food. Moreover, complete unanimity was verified concerning the restrictions on the use of pastoral lands, since all respondents answered "Yes" to the question "Does the Park limit livestock activities?".

To promote sustainable tourism in Iona National Park, it is necessary to achieve a solid compromise between nature conservation goals and satisfaction of the needs and well-being of the local population. However, respondents' answers concerning positive and negative effects of the Park revealed that such a compromise may not be easy to reach. On the one hand, the local population recognizes that the Park has brought, or is expected to bring in the near future, significant benefits. On the other hand, most respondents do not seem prepared to abandon an extensive use of natural resources for the sake of nature conservation. To reconcile these conflicting perspectives, tourism policies must be grounded in inventories and projections of resource use as well as in the predictions of interlocking resource relationships. Multiple uses of pastoral land, which is a cardinal resource for both wildlife and livestock, should be one of the main objectives prioritized throughout the study area. Currently, both hunting tourism and meat hunting by the local population are prohibited. This hunting ban is entirely justified because game species densities do not allow a partial removal of the population. However, once these species reach the environmental carrying capacity, meat hunting by indigenous people should be equated. Allowing legal hunting may avoid indiscriminate killing in defense of human life and livestock, as well as prey depletion due to poaching and bushmeat trade. Controlled

hunting of game species by the local population may help in overcoming human–wildlife conflicts and support nature conservation.

Other conflicting interests should be addressed and managed according to the locality's profile, separately treating the more urbanized areas located in the heart of the Park, such as Iona, and the deeply rural areas, such as Ngulova. Knowing villagers' different perspectives will also help to find ad hoc resolutions for specific environmental challenges, such as adjusting indigenous resource needs to the uses of different areas according to the established zoning scheme. The knowledge of particular characteristics held by distinctive subgroups of villagers may also be useful in matters of human resource management, for example, helping the Park Administration find the right hires among the indigenous people. Knowing villagers' different perceptions concerning the negative effects of certain tourism activities may be of help in the prevention of social, behavioral, and psychological impacts. Using this knowledge, an ordinary tourist may end up behaving similar to real ecotourist. For instance, in localities where people are more negatively affected by tourism activities, such as Iona, tourists should not travel independently but under the guidance of an accredited operator. The same knowledge may also be useful in prioritizing the recipients of a measure or action. For example, if the Angolan government decides to construct a road connecting the Iona National Park to the neighboring Namibian Skeleton Coast Park, people from Ngulova, who are mostly against such an infrastructure, should be the first target of a persuasion campaign. Planning for sustainable nature-based tourism and ecotourism in protected areas requires one main prerequisite—the full consideration of stakeholder interests, namely local community interests, in managing the protected area [35]. We hope that this study may significantly contribute to achieving this goal in Iona National Park.

**Author Contributions:** Field work and data processing were carried out by J.M. under the supervision of P.S. and C.P.-G. The results were analyzed and interpreted by J.M., R.C., and P.S. The original draft was written by J.M. and the final paper was written by J.M. and P.S in collaboration with all co-authors.

**Funding:** This research received no external funding.

**Acknowledgments:** The authors gratefully acknowledge R.M. for his graphic assistance. We thank Instituto de Ciências Agrárias e Ambientais Mediterrânicas (ICAAM) for editing services.

**Conflicts of Interest:** The authors declare no conflict of interest.

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
