# Peer review of "Villagers’ Perceptions of Tourism Activities in Iona National Park: Locality as a Key Factor in Planning for Sustainability"

_sustainability, doi:10.3390/su11164448_

Round 1

Reviewer 1 Report

It is a very interest theme with high originality and very well presented by the authors. 

The only weak point of this research is if the results of the present survey, because of the the low educational level of the villagers are reliable because I think that the most of the surveyed people give the answers yes or no  in a rather stress condition.In the questionnaire could be included and questions with answers without only yes or no.

Improvements to the manuscript:

In unit 4.2 page 13  in the title the word of should be deleted.

Also in unit 4.5 page 15  in the title in the near future could be change to in the near future

Author Response

We thank you for the kind words and valuable suggestions.

We followed your suggestions, the text was reviewed by a native speaker (MDPI English pre-edit services) and we believe that the manuscript has significantly improved. 

"It is a very interest theme with high originality and very well presented by the authors. 

The only weak point of this research is if the results of the present survey, because of the low educational level of the villagers are reliable because I think that the most of the surveyed people give the answers yes or no  in a rather stress condition. In the questionnaire could be included and questions with answers without only yes or no."

We deliberately chose closed-ended questions because of the expected communication problems, mainly because many respondents had difficulty with speaking Portuguese. Some respondents were only proficient in their own geographic dialect. Additionally, closed-ended questions facilitated the categorisation of respondents and allowed us to conduct statistically significant tests. Lines 266-270.

"Improvements to the manuscript:

In unit 4.2 page 13  in the title the word of should be deleted."

The word “of” was deleted. Line 424.

"Also in unit 4.5 page 15  in the title in the near future could be change to in the near future"

The title was changed according to the suggestion. Line 539.

Reviewer 2 Report

I read the manuscript entitled »Villagers’ perceptions of tourism activities in Iona National Park: Contributing to improve tourism management towards sustainability«. The authors studied local people's perception of tourism activities in Angola and their attitudes toward present and prospective effects of tourists on local population's livelihoods and cultural integrity. The authors administered a questionnaire with help of park rangers. Altogether they managed to collect 202 responses from various ethnic groups dwelling in or near the Iona National Park, Angola. The aim of the paper was to help the Park authorities in planning future management activities related to ecotourism and to improve natural resource governance in similar conditions.

The paper is short of general significance or at least the authors did not manage to go beyond simple description of the results. Many questions in this kind of attitudinal studies are only indicator variables measuring attitudes toward a latent construct, e.g. attitudes toward ecotourism. A reader is probably not interested in an answer to a specific question (yes, no, %) but more so in attitudes toward certain aspects of ecoturism, e.g. financial aspects, organizational aspects, cultural problems that tourism might ignite etc. Dimension reduction methods such as Principal Component Analysis can reduce the number of dimensions and point to specific problems/aspects which respondents are actually communicating through answering individual questions. In any case, the results should be presented in a more synthetic way, now the analytical approach prevails.

This is mainly because there are no specific hypotheses presented at the beginning of the paper and most of the text remains descriptive and atheoretical. Be more elaborative in your starting points and think about which general recommendations your study supposed to bring for ecotourism (except for the well-known recommendation that local interests should be taken into account). You merely compare the associations between age, locality, ethnicity, gender. What for?

Most importantly, I was confused about the people’s opinion. On the one hand, local population seems to support ecotourism. No one disagreed with the statements in Table 3, which measure direct positive effects of tourism on local communities. On the other hand, in the next set of questions on the negative effects (Table 4) all unanimously agreed on the negative effects of the Iona Park. These answers probably indicate socially-desirable behavior or other kind of acquiescence. Constant agreement or disagreement may be linked to several external and internal stimuli.  Inter alia, it may depend on an individual’s risk attitudes, it may be influenced by social norms, it may be related to the demographic variables and personality characteristics of a respondent. A lack of interest in the topic (“yeah answers”) may also lead to bias.The results are really contradictory. Maybe the reason for this are the interviewers -  Park guards. Any commentary on that?

Please add a paragraph on the Iona Park, where you describe the history of the Park, management regime, previous conflicts and challenges in Park governance, the impact of the Angolan Civil War and illegal poaching. Please mention and discuss any other studies or projects on the development of this area. The Park has been proclaimed National park back in 1960s and the context of your study is not quite clear. What was the reason for doing the survey? Are there massive efforts for a change in the paradigm toward ecotourism also in the broader region?

You discuss hunting and possible revenues that hunting may bring to local population, but how does this goes hand in hand with hunting prohibition in the Park? What is your recommendation for hunting regime in the Park? Do villagers have their own hunting privileges?

I believe one of the points of the paper was to gear up the government with necessary knowledge on the local people’s perception of the Park to improve governance. What is then the message for decision-makers? The conclusions are very general and not really novel. For instance, lines 504-506 “To promote sustainable tourism in Iona National Park, it is necessary to achieve a solid compromise between nature conservation goals and satisfaction of needs and well-being of local population” have been seen so many times…

Other comments:

The title is too complicated. Had the results of the survey been clear, the message could be clearly stated in the title, like whether or not people support or oppose the Park, or where the biggest pitfalls for the successful operation of the Park are.

l. 25: “present and prospective positive effects of the Park; present and 25 prospective negative effects of the Park” can be merged into “present and prospective positive and negative effects of the Park”.

l. 29: “noticed” is redundant.

Keyword “organisations and tourists behaviour changes” should be replaced with something else.

l. 36: Change to: “Besides having well-known negative ecological impacts…”

There are minor grammatical errors in the text, I suggest a review by a native speaker. E.g., l. 18 impacts on, l. 25 of and of, l. 52 ”though”?, l. 108, l. 180 kraal?, l. 246, l. 330 between parentheses, in parentheses, l. 349, l. 357, l. 436, l. 455, l. 476…

l. 207-208: “Among this people, each person remaining in the population had equal chance of being drawn during each selection round.” Self-selection sampling is not probability sampling. This statement is incorrect.

What is the purpose of doing the analysis per ethnicities? What was your assumption?

l. 283-285: This is an example of a sentence with no clear significance for the study; it is just a boring description of frequencies. Did you have any expectations on how the people from Pediva will answer your questions?

Table 4: “increase” should be “increased”

l. 335-337: “Our sample includes a disproportional number of males and a disproportional number of villagers belonging to the age-class 41-50 years”. How do you know that?? You have no data on the population level, do you?

l. 425: “however, since Gender and Ethnic group are not independent from Locality, we focused the results analysis on Locality and Age, the only biodata variable which is independent from Locality.” Locality is independent from Locality?

Author Response

We thank you for the valuable remarks and suggestions.

We have revised the manuscript according to every comment you made, and we believe the version now submitted incorporates very significant improvements. 

To address the issues “The paper is short of general significance …”, “… most of the text remains descriptive and atheoretical”, “… there are no specific hypotheses presented at the beginning of the paper…” and “Be more elaborative in your starting points and think which general recommendations your study supposed to bring for ecotourism”, we provided more information to the reader on how the study will contribute, what the study will contribute and who will benefit from it by including in the Introduction the following text:

We hypothesized that locality would be the main factor that affected villagers’ perceptions because human settlements in the Park present strong differences concerning level of urbanisation and tourism development, and they are also very divergent with regard to elements of tourism attractiveness, both biotic and abiotic. We also analysed whether age, gender, and ethnic group, demographic factors which are known to affect peoples’ attitudes and opinions, influenced villagers’ perceptions. In synthesis, the major goal of the present study is to provide Park authorities with information that may be useful to attain a sound compromise between two main principles of ecotourism: the well-being of the local population, and nature conservation. To make it easier for authorities to reach their recipients, we focused our analysis in finding differences between demographic profiles. Lines 80-89.

We believe this information also answers your question “What for?”, regarding the comparisons we make between age, locality, ethnicity and gender.

Also in the Introduction, in order to emphasize the current relevance of the study and following your recommendation to “.. add a paragraph on the Iona Park, where you describe the history of the Park, management regime, previous conflicts and challenges in Park governance, the impact of the Angolan Civil War and illegal poaching. Please mention and discuss any other studies or projects on the development of this area. The Park has been proclaimed National park back in 1960s and the context of your study is not quite clear. What was the reason for doing the survey? Are there massive efforts for a change in the paradigm toward ecotourism also in the broader region?” we rewrote the previous text adding abundant complementary information. The text now reads: 

Iona National Park, which is the oldest and largest Protected Area in Angola, covers around 15.150 km2 and is located in the Namibe Province [11]. Once a Game Reserve, established in 1937, it was proclaimed a National Park in 1964 still under the Portuguese Administration. The Angolan Civil War (1975–2002) led to a long period of abandonment, resulting in devastating consequences for the wildlife. Several emblematic species became Critically Endangered or even Extinct at the regional level, such as the black rhinoceros (Diceros bicornis), the African wild dog (Lycaon pictus), the black-faced impala (Aepyceros melampus petersi), the mountain zebra (Equus zebra), the African buffalo (Syncerus caffer), and the cheetah (Acinonyx jubatus). To reverse this situation, several organizations across Southern Africa have recently joined efforts, such as is the case of The Range-Wide Conservation Program for Cheetah and African Wild Dogs. To preserve a continuous block of the Namib Desert coastline and adjacent dunes, governmental authorities of Angola and Namibia may work together in order to create one of the largest transboundary conservation and tourism areas in Africa. Under the auspices of the United Nations, an Angolan National Project (UNDP Project ID 4082) aimed at conserving the country’s biodiversity, beginning with the conservation of Iona National Park, was developed between February 2013 and April 2018. Lines 108-122.

“Many questions in this kind of attitudinal studies are only indicator variables measuring attitudes toward a latent construct, e.g. attitudes toward ecotourism. A reader is probably not interested in an answer to a specific question (yes, no, %) but more so in attitudes toward certain aspects of ecoturism, e.g. financial aspects, organizational aspects, cultural problems that tourism might ignite etc.”

“These answers probably indicate socially-desirable behavior or other kind of acquiescence. Constant agreement or disagreement may be linked to several external and internal stimuli.  Inter alia, it may depend on an individual’s risk attitudes, it may be influenced by social norms, it may be related to the demographic variables and personality characteristics of a respondent. A lack of interest in the topic (“yeah answers”) may also lead to bias.The results are really contradictory. Maybe the reason for this are the interviewers -  Park guards. Any commentary on that?”

Most of our respondents are not familiar with concepts like ecotourism and, thus, they don’t have words to express them. So, we must admit that probably many of the yes/no answers they gave were influenced by how the questions were presented to them by the Park guards and by the interpreters. This drawback reflects the lack of conditions to present the villagers open questions. To justify our decision to use closed-ended questions we wrote the following text:

We deliberately chose closed-ended questions because of the expected communication problems, mainly because many respondents had difficulty with speaking Portuguese. Some respondents were only proficient in their own geographic dialect. Additionally, closed-ended questions facilitated the categorisation of respondents and allowed us to conduct statistically significant tests. Lines 266-270.

“Dimension reduction methods such as Principal Component Analysis can reduce the number of dimensions and point to specific problems/aspects which respondents are actually communicating through answering individual questions. In any case, the results should be presented in a more synthetic way, now the analytical approach prevails.”

We agree that the results presentation could be improved by using a method, such as PCA, that would allow the detection and graphic representation of the major underlying structures in our dataset. As we collected data in an ordinal mode (all our variables are qualitative), instead of PCA we used multiple correspondence analysis (MCA). In the Results, adding Figures 3 and 4 (pages 14 and 15), we wrote the following text:

The graphic results of the Multiple Correspondence Analysis (MCA) used to uncover the relationships among the studied categorical variables are presented in Figures 3 and 4. The projection of respondents on the first plane, presented in Figure 3, showed that the respondents within the same Locality were close together. The discrimination measures of the variables are exhibited in Figure 4, which revealed that Locality presented the highest values in both dimensions. Along Dimension 1, which is represented by the horizontal axis and accounts for 24.74% of the variance, we see in Figure 4 that respondents from Iona (e.g., object points 116, 133, 134 and 161) and respondents from Ngulova (e.g., object points 72, 75, 81, 86, and 93) were furthest away from the origin and, therefore, had the most importance. Besides Locality, the variables referring to the following questions also had high discrimination values: Does the Park contribute to the breakdown of family relationships or within the community?; Does the Park contribute to the disturbance of the tranquillity of the populations and/or their cultural manifestations?; Does the Park contribute to the construction of infrastructures not wanted by the population?; Does the Park contribute to increased garbage made by tourists?. Thus, the divergence of Iona and Ngulova mainly was due to differences between these localities regarding villagers’ perceptions concerning the negative aspects of the Park. Along Dimension 2, which is represented by the vertical axis and accounts for 12.84% of the variance, we see that respondents from Garotas Novas (e.g., object points 6, 7, 12, 14, 15, and 16) and respondents from Espinheira (e.g., object 24, 25, 26, 27, 28, and 29) had the most importance. Besides Locality, only the variable referring to the question Do tourists fish? presented a high value. These results indicate that the largest deviation from independence in the sample was between Iona/Ngulova and seemed to result mainly from differences regarding level of urbanization and tourism development. The second most important difference was between Garotas Novas/Espinheira and seemed to result mainly from differences regarding tourism facilities and elements of attractiveness. All respondents from Curoca are close to the origin, which means that they tend to be neutral regarding both dimensions. Lines 355-378.

The use of MCA allowed us to present the results in a more synthetic way and also to better support our conclusions. However, the two first Dimensions left to explain a large proportion of the variation, and further Dimensions explain increasingly smaller parts of the variance (for instance the fifth Dimension accounts for only about 7% of the variance). On the other hand, MCA does not allow to test if the found relationships are statistically significant. Moreover, we wanted to analyse if there were statistically significant differences between demographic profiles. Thus in the Discussion we wrote the following text:

The two first Components or Dimensions account for about 40% of the variance. These results suggest the use of an analytical approach to break down the complex system of relationships into smaller pieces. Herein, we analyse in detail some of those small pieces by comparing villagers’ perceptions between the different classes of Age, Gender, Ethnic group, and Locality. We also analyse whether these demographic variables are independent of each other. Finding statistically significant differences between different demographic profiles would make it easier for the Park authorities to reach their intended recipients and, thus, to implement the desired measures and actions. Lines 393-399.

“Most importantly, I was confused about the people’s opinion. On the one hand, local population seems to support ecotourism. No one disagreed with the statements in Table 3, which measure direct positive effects of tourism on local communities. On the other hand, in the next set of questions on the negative effects (Table 4) all unanimously agreed on the negative effects of the Iona Park.”

“You discuss hunting and possible revenues that hunting may bring to local population, but how does this goes hand in hand with hunting prohibition in the Park? What is your recommendation for hunting regime in the Park? Do villagers have their own hunting privileges?”

We addressed both these issues in the Conclusion by rewriting the text, now it reads:

To promote sustainable tourism in Iona National Park, it is necessary to achieve a solid compromise between nature conservation goals and satisfaction of the needs and well-being of the local population. However, respondents’ answers concerning positive and negative effects of the Park revealed that such a compromise may not be that easy to reach. On the one hand, the local population recognises that the Park has brought, or is expected to bring in the near future, significant benefits. On the other hand, most respondents do not seem prepared to abandon an extensive use of natural resources for the sake of nature conservation. To reconcile these conflicting perceptions, tourism policies must be grounded in inventories and projections of resource use as well as in the prediction of interlocking resource relationships. Multiple uses of pastoral land, which is a cardinal resource for both wildlife and livestock, should be one of the main objectives prioritized throughout all the study area. Currently, both hunting tourism and meat hunting by the local population are prohibited. This hunting ban is entirely justified because game species densities do not allow a partial removal of the population. However, once these species reach the environmental carrying capacity, meat hunting by indigenous people should be equated. To allow legal hunting may avoid indiscriminate killing in defense of human life and livestock as well as prey depletion due to poaching and bushmeat trade. Controlled hunting of game species by the local population may help in overcoming human–wildlife conflicts and concur with nature conservation. Lines 578-594.

“I believe one of the points of the paper was to gear up the government with necessary knowledge on the local people’s perception of the Park to improve governance. What is then the message for decision-makers? The conclusions are very general and not really novel. For instance, lines 504-506 ‘To promote sustainable tourism in Iona National Park, it is necessary to achieve a solid compromise between nature conservation goals and satisfaction of needs and well-being of local population” have been seen so many times…’ ”

The message for the decision-makers is synthesised in the Conclusion. We agree that in the previous version of manuscript some of the conclusion were general guidelines trying to match ecotourism principles. In the current version, besides rewriting a large part of the Conclusion adding information regarding specifically the Iona Park (see the text above), we also provide some concrete examples (see text below) of how the Park authorities may use the produced information to promote sustainable tourism.

For instance, in localities where people are more negatively affected by tourism activities, such as Iona, tourists should not travel independently but under the guidance of an accredited operator. The same knowledge may also be useful for prioritizing the recipients of a measure or action. For example, if the Angolan government decides to construct a road connecting the Iona National Park to neighboring Namibian Skeleton Coast Park, people from Ngulova, who are mostly against such an infrastructure, should be the first target of a persuasion campaign. Lines 605-610.

“Other comments:

The title is too complicated. Had the results of the survey been clear, the message could be clearly stated in the title, like whether or not people support or oppose the Park, or where the biggest pitfalls for the successful operation of the Park are.”

We partially changed the title by replacing and abstract sentence “Contributing to improve tourism management towards sustainability” with the study major conclusion “Locality as a key factor in planning for sustainability”.

“l. 25: ‘present and prospective positive effects of the Park; present and 25 prospective negative effects of the Park’ can be merged into ‘present and prospective positive and negative effects of the Park’.

We followed the suggestion.

“l. 29: ‘noticed’ is redundant”

We deleted the word.

“Keyword ‘organisations and tourists behaviour changes’ should be replaced with something else.”

We replaced “organisations and tourists behaviour changes” with “destination management”.

“l. 36: Change to: ‘Besides having well-known negative ecological impacts…’”

We changed the text according to the recommendation.

“There are minor grammatical errors in the text, I suggest a review by a native speaker. E.g., l. 18 impacts on, l. 25 of and of, l. 52”though”?, l. 108, l. 180 kraal?, l. 246, l. 330 between parentheses, in parentheses, l. 349, l. 357, l. 436, l. 455, l. 476…”

We corrected the pointed out grammatical errors and, as suggested, the text was reviewed by a native speaker (MDPI English pre-edit services).

“l. 207-208: ‘Among this people, each person remaining in the population had equal chance of being drawn during each selection round.’ Self-selection sampling is not probability sampling. This statement is incorrect.”

We agree with the comment and deleted the statement.

“What is the purpose of doing the analysis per ethnicities? What was your assumption?”

We answered this question in the Introduction by writing:

We also analysed whether age, gender, and ethnic group, demographic factors which are known to affect peoples’ attitudes and opinions, influenced villagers’ perceptions. Lines 83-85.

“l. 283-285: This is an example of a sentence with no clear significance for the study; it is just a boring description of frequencies. Did you have any expectations on how the people from Pediva will answer your questions?”

We agree with the comment and deleted the sentence.

“Table 4: ‘increase’ should be ‘increased’ ”

We replaced “increase” with “increased” in Table 4 and along the text.

“l. 335-337: ‘Our sample includes a disproportional number of males and a disproportional number of villagers belonging to the age-class 41-50 years’. How do you know that?? You have no data on the population level, do you?”

We rewrote the text, it now reads:

Assuming that the demographic structure of the studied population is similar to that presented by the Angolan total population, our sample included a disproportionate number of males and a disproportionate number of villagers belonging to the age-class 41–50 years (see Table 1). Lines 403-406.

“l. 425: “however, since Gender and Ethnic group are not independent from Locality, we focused the results analysis on Locality and Age, the only biodata variable which is independent from Locality.” Locality is independent from Locality?”

We rewrote the text, it now reads:

The results varied according to the respondents’ demographic profile; however, since Gender and Ethnic group were not independent from Locality, we focused the analysis on Locality and Age, as the latter was the only demographic variable that was independent from Locality. Lines 495-497.

Round 2

Reviewer 2 Report

The manuscript has been significantly improved; I like the title much more.

I appreciate the authors conducted additional analyses and paid attention to a synthetic type of writing, particularly in the discussion/conclusions.